# Personalized tumor-specific DNA junctions to detect circulating tumor in patients with endometrial cancer

Tommaso Grassi[1,2☯], Faye R. Harris[3☯], James B. Smadbeck[3], Stephen J. Murphy[3], Matthew S. Block[4], Francesco Multinu[1,5], Janet L. Schaefer Klein[3], Piyan Zhang[3], Giannoula Karagouga[3], Minetta C. Liu[4], Alyssa Larish[1], Maureen A. Lemens[1], Marla Kay S. Sommerfield[1], Serena Cappuccio[1,6], John C. Cheville[7], George Vasmatzis[3‡]*, Andrea Mariani[1‡]

1 Department of Obstetrics and Gynecology, Mayo Clinic, Rochester, MN, United States of America, 2 Clinic of Obstetrics and Gynecology, University of Milan-Bicocca, San Gerardo Hospital, Monza, Italy, 3 Center for Individualized Medicine, Mayo Clinic, Rochester, MN, United States of America, 4 Department of Oncology, Mayo Clinic, Rochester, MN, United States of America, 5 Department of Gynecology, IEO, European Institute of Oncology IRCCS, Milan, Italy, 6 Department of Women and Child Health, Catholic University of the Sacred Heart, Rome, Italy, 7 Laboratory Medicine and Pathology, Mayo Clinic, Rochester, MN, United States of America

☯ These authors contributed equally to this work.
‡ These authors are joint senior authors on this work.
* Vasmatzis.george@mayo.edu

**Data Availability Statement:** For this specific study, our current IRB protocol and consent form did not allow us to deposit genomic data to a public repository because it is considered health information, and we did not get the explicit

## Abstract

### Introduction

There are no reliable blood biomarkers for monitoring endometrial cancer patients in the current clinical practice. Circulating tumor DNA (ctDNA) is emerging as a promising non-invasive method to measure tumor burden, define prognosis and monitor disease status in many solid cancers. In this pilot study, we investigated if unique tumor-specific DNA junctions can be used to detect ctDNA levels in patients with endometrial cancer.

### Methods

Chromosomal rearrangements in primary tumors of eleven patients with high-grade or advanced stage endometrial cancer were determined by whole-genome Mate-Pair sequencing. Identified unique tumor-specific junctions were evaluated in pre- and six-week post-surgery patient plasma using individualized quantitative polymerase chain reaction (qPCR) assays. The relationship between clinicopathological features and detection of ctDNA was investigated.

### Results

CtDNA was detected in 60% (6/10) of cases pre-surgery and in 27% (3/11) post-surgery. The detection of ctDNA pre-surgery was consistent with clinical indicators of aggressive disease such as advanced stage (80% - 4/5), lymphatic spread of disease (100% - 3/3), serous histology (80% - 4/5), deep myometrial invasion (100% - 3/3), lympho-vascular space invasion (75% - 3/4). All patients in which ctDNA was detected post-surgically had type II endometrial cancer.

permission from the patients to do so. To make some of the MPseq data available without compromising patient privacy concerns, we have added two supplemental tables: S2 Table which includes a list of all junctions detected for each case (discordant genome as compared to reference genome), and S3 Table which includes a report of all supporting reads for each junction. Note, data supporting concordant sequencing (which aligns with the reference genome and may contain low coverage information on SNPs) is included. Investigators who would like access to this data are invited to email the the Research Compliance Office (RCO@mayo.edu) with a data request, which will be considered on a case-by-case basis in accordance with our IRB and general Mayo Clinic policies regarding sharing data.

**Funding:** This work was supported by the Center for Individualized Medicine, and the Biomarker Discovery Program within the Center for Individualized Medicine, both of Mayo Clinic. The funders had no role in study design, data collection and analysis, decision to publish, or preparation of the manuscript.

**Competing interests:** The authors declare no competing financial interests.

## Discussion

This pilot study demonstrates the feasibility of using personalized tumor-specific junction panels for detecting ctDNA in the plasma of endometrial cancer patients. Larger studies and longer follow-up are needed to validate the potential association between pre-surgical ctDNA detection and the presence of cancers with aggressive pathologic tumor characteristics or advanced stage observed in this study.

## Introduction

Endometrial cancer is the most common malignancy of the female genital tract in the United States. While frequently diagnosed at an early stage and with a relatively good prognosis, its mortality rate has been increasing during the last 3 decades [1]. In approximately 20% of patients with an apparent early endometrial cancer, occult extra-uterine disease and lymphatic involvement is ascertained at the time of surgery. Currently no pre-operative imaging tests or blood biomarkers are sensitive enough to detect microscopic extra-uterine spread of disease [2]. Therefore, surgical staging, including lymph node status evaluation, is recommended to identify patients at high-risk of recurrence and tailor the adjuvant treatment [3]. Additionally, no reliable blood-based biomarkers are available to monitor treatment responses or recurrence during follow-up.

Cell-free circulating tumor DNA (ctDNA) is emerging as a promising non-invasive method to measure tumor burden, define prognosis and monitor disease status in many solid cancers [4–8]. Total plasma cell-free DNA (cfDNA) levels in endometrial cancer patients were reported to be increased compared to healthy controls [9, 10], but lacks specificity to robustly discriminate pathological levels versus benign controls. A few studies have evaluated the feasibility of using tumor-specific mutations to detect ctDNA in the plasma of endometrial cancer patients [11–16]. In particular, ctDNA screening for point mutations present in primary gynecological tumors using digital droplet PCR reported promise for detecting recurrence preceding anatomic findings on standard computed tomography scans by an average of 7 months [11]. However, as the majority of cases were ovarian cancers, conclusions are limited regarding endometrial cancer and more studies are warranted.

Large genomic rearrangements commonly occur in solid tumors, resulting in highly unique somatic DNA junctions from chromosomal shuffling's. These DNA junctions present as alternatives to point mutations as unique tumor-specific DNA markers for ctDNA monitoring. Genome wide DNA junctions can be accurately identified at the precise breakpoint level by mapping fragments discordant to the reference human genome as obtained from Mate-Pair sequencing of tumor derived DNA [17, 18].

The utility of DNA junctions as unique highly specific markers of ctDNA levels in ovarian cancer patients was previously demonstrated [8]. In this pilot proof-of-concept study, we sought to determine whether the pre- and post-surgical ctDNA status is associated with the extent and clinicopathological features of the disease.

## Materials and methods

### Patient selection

Eleven patients with suspected high-grade endometrial cancer were prospectively enrolled at Mayo Clinic, Rochester, MN from 7/8/2017 to-11/8/2017. For all the patients we had tumor tissue and plasma collected. All patients underwent hysterectomy (± bilateral salpingo-oophorectomy) and lymph node evaluation. No patient received chemotherapy or radiation therapy

pre-surgery. CA125 was measured within 30 days pre-surgery. The histologic subtypes and grades of the tumors were evaluated according to WHO criteria [19]. All patients provided written informed consent for the use of their tissue, blood, and electronic medical record data. The study protocol was approved by the research ethics committee Institutional Review Board of Mayo Clinic.

## Tumor tissue DNA isolation and next-generation sequencing

DNA isolated from fresh-frozen macrodissected primary tumor tissue was sequenced using the whole genome Mate-Pair sequencing protocol and analyzed to detect structural variants. DNA was isolated from macrodissected primary tumor tissue using the Qiagen AllPrep DNA/ RNA mini kit (Qiagen, #80204) or Qiagen DNeasy Blood and Tissue Kit (Qiagen, #69504) for peripheral blood mononuclear cell (PBMC) extraction following the manufacture's protocol. The whole-genome mate-pair sequencing (MPseq) protocol was utilized to detect structural variants at the base level resolution through specialized larger 2-5kb fragment tiling of the genome [17, 18]. One microgram of DNA was applied to mate-pair library preparation using the Nextera Mate-Pair Kit (Illumina, #FC-132-1001) following the manufacturer's protocol. Libraries were sequenced on the Illumina HiSeq4000 platform at a depth of four libraries per lane.

The binary indexing mapping algorithm (BIMA), developed by the Biomarker Discovery Lab at Mayo Clinic specifically for MPseq data, simultaneously maps both reads in a fragment to the GRCh38 reference genome [20]. Structural variants are detected using SVAtools, a suite of algorithms also developed by the Biomarker Discovery Lab at Mayo Clinic, using a minimum of three supporting reads for junction detection [17]. One to five unique somatic DNA junctions per patient were selected from each patient tumor for plasma screening according to the following criteria: 1) Larger numbers of supporting-reads prioritized for increased confidence and potential higher tumor representation. 2) Junctions linked to replicating copy gains for increased sensitivity of detection. 3) Junctions sequenced through the precise breakpoints, enabling optimal primer design. 4) Low potential of being germline. 5) Avoiding junctions in highly repetitive regions of the genome to minimize potential off-target signals. 6) Ability to design primers with an amplicon size of ~100bp compatible with cfDNA fragments (~146-166bp). CNV detection is performed using the read count of concordant fragments within non-overlapping bins (S1 Table) [18]. This algorithm uses both a sliding window statistical method to determine likely copy number edges from read depth, as well as using breakpoint locations determined in the junction detection stage to more accurately place these edges. Once the genome is segmented into likely copy number regions, the normalized read depth for a region is calculated as two times the read depth within a region divided by the expected read depth for normal diploid level for the sample. Chromosomal copy levels and discordant mapping junctions are visualized on interactive software for genome U-plots (S1 Fig) [21]. Junctions were selected to enable greater specificity of detection rather than SNVs, which are more susceptible to false positives due to polymerase missense mistakes [22]. Full MPseq data may be available at the discretion of Mayo Clinic genomic data policies and the Mayo Clinic IRB by contacting the corresponding author. A table of all detected junctions of each case is available (S2 Table). Additionally, all reads supporting each junction, defined as at least a 30kb difference in genomic position for intrachromosomal reads, or interchromosomal mate pairs, is available (S3 Table)

## Plasma processing and cfDNA isolation

Pre- and post-surgical plasma were prospectively collected within a week pre-surgery and 6-weeks post-surgery (Fig 1A). Pre- and post-surgical plasma was available for all patients but

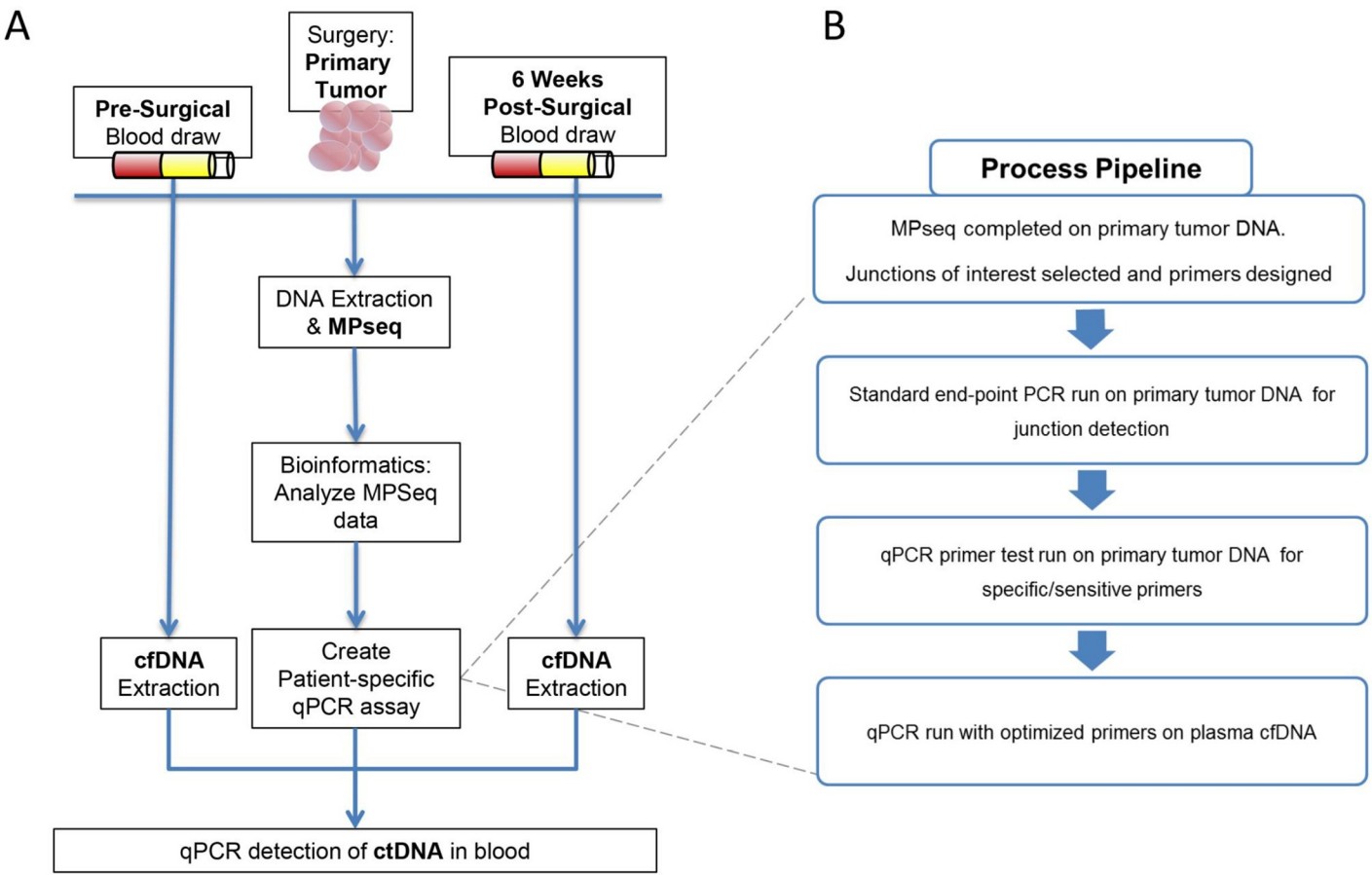

**Fig 1. Study design and process pipeline for circulating tumor DNA (ctDNA) detection.** **(A)** Blood is drawn pre–and six weeks post–surgery in patients with endometrial cancer. DNA from the tumor is sequenced using the mate pair sequencing protocol. Junctions are detected and used to design individualized ctDNA qPCR assays to interrogate the blood for the presence of tumor DNA. **(B)** Primers are designed for selected junctions and undergo a pipeline to test for sufficient specificity and sensitivity to be used in the ctDNA assays.

one; CTE029, where only post-surgical blood was available. Platelet-poor plasma was obtained from whole blood drawn into two 10mL STRECK cell-free DNA BCT tubes (STRECK, #218962). The plasma was separated from the buffy coat within 4 hours of collection and centrifuged for 10 minutes at 2k rpm. The plasma was then spun again at 2k rpm for 10 minutes to yield platelet-poor plasma and stored at -80˚C until cfDNA extraction. CfDNA was isolated from 1-5mL of platelet-poor plasma using the Circulating Nucleic Acid kit and eluted into 10-30ul of elution buffer (Qiagen, MD, USA; 55114), either following the manufacturer's protocol or with the following modifications: columns were eluted twice with 150ul elution buffer heated to 37˚C. 140ul of each elution was combined and concentrated to 10-30ul using the DNA clean and concentrator-5 kit (Zymo, #DCC-5). CfDNA yield was quantitated using the Qubit dsDNA HS assay kit (Thermo Fisher Scientific, #Q32854) (S4 Table). Normal control cfDNA (NC cfDNA) from pooled EDTA plasma of cancer-free individuals was processed following the manufacturer's protocol.

## Individualized monitoring panels for ctDNA detection

All DNA junctions detected in each patient were evaluated using the criteria described above and one to five unique junctions were initially selected for potential plasma testing (S2–S4

Tables). Primers flanking each junction, designed to yield product sizes of <100bp, were first optimized by standard end-point PCR on patient tumor DNA, which had been whole-genome genome amplified (Qiagen RepliG #150025) to conserve primary tumor DNA. Independent pooled human genomic DNA (gDNA) (Promega, #G3041 for standard PCR or Life Technologies, #4312660 for qPCR) and normal patient control cfDNA (NC cfDNA) were used as negative controls. Promising primer combinations were next assessed by qPCR on serial diluted patient tumor DNA (Fig 1B). These serial dilutions were primarily used as an initial evaluation of linearity, robustness of duplicates, and used for ctDNA calculations (S2 Fig). An $R^2$ of at least 0.90 was required for sufficient robustness, as well as demonstrating high specificity. Selected primer pairs required absences of significant non-specific signals in gDNA, NC cfDNA or water controls. Junctions were additionally confirmed somatic through a standard end-point PCR on patient PBMC derived germline DNA. Final qPCR screening for ctDNA were performed using selected primers (150nM) (S5 and S6 Tables), 2x Sybr Green (Invitrogen, #4367659) and patient cfDNA (3ul) or controls in a total reaction volume of 12ul (S4 Table). Higher sensitivity semi-nested PCR was performed to further verify undetected ctDNA results in presurgical samples, using a pre-amp standard end-point PCR (30 cycles) using overlapping primer sets with different 5' or 3' priming positions. Primary PCR reactions were cleaned using Kappa beads (Roche #7983271001) with the secondary nested qPCR performed as outlined above.

The Applied Biosystems ViiA-7 instrument was used for all qPCRs using a standard protocol of 40 cycles of denaturing (95˚C for 15 seconds) and customized annealing for each panel (55˚C-62˚C for 1 minute). Standard curves of serially diluted tumor DNA were used to assess linearity (S2 Fig) and melting curves were analyzed for specificity. Primers amplifying NAGK were used as an internal control for total cfDNA [23].

### Quantification of ctDNA

The quantity of ctDNA was calculated using the algorithm previously described [8] with the following modifications: The scaling factor [Chi] that quantified the fraction of the tumor cells with the junction present was assumed to be 1 in all cases as we exclusively chose junctions likely present in all tumor cells. The experimentally derived cycle thresholds (Ct) of ctDNA detected used in the calculation, can be found in S7 Table. In cases where multiple junctions were detected in the blood, the junction with the lowest Ct was selected for use in the calculation as it was considered the most sensitive measurement of ctDNA.

## Results

### Patient demographics

Clinical and pathologic characteristics for the patients are summarized in Table 1 and S8 Table. Stage I disease was diagnosed in 45% (5/11) of patients and advanced stage disease (stage III-IV) in the remaining 55% (6/11). Approximately 27% (3/11) had endometrioid histology while 73% (8/11) had type 2 endometrial cancer: 5 serous, 2 carcinosarcoma and 1 mixed histology (serous/endometrioid). Ninety percent of the cases were grade 3 (10/11). Deep myometrial invasion was observed in four of five stage III tumors, ranging from 72–100%, with the additional serous IIIC2 tumor presenting with zero myometrium invasion. In stage 1A tumors the myometrial invasion ranged from 23–42%. The only stage IVB serous tumor had a 21% myometrial invasion and negative follow-up ranged from 4 to 44 months post-surgery. Five patients experienced recurrences or progression of disease, three of whom died of disease (Table 1).

**Table 1. Clinical characteristics of the endometrial cancer patients and circulating tumor DNA detection.**

| Patient | Histology | FIGO Stage | Grade | Myometrial invasion (%) | Lymph-Vascular Space Invasion | Lymph Node Status | SLN vs. LND (N of nodes) | Omental / Peritoneal disease | CA125 pre-Surgery | Adjuvant treatment | Time of FUP (months) | Vital status | Time (months) to Relapse (R) or Progressive Disease (PD) | Sites of Relapse or Progressive Disease |
|---|---|---|---|---|---|---|---|---|---|---|---|---|---|---|
| CTE006 | Endom | IA | 3 | 23% | Yes | Neg | SLN (4) | No | 50 | VRT | 44 | AWD | 32 (R) | Peritoneum; Liver |
| CTE034 | Endom | IA | 3 | 29% | No | Neg | SLN (4) | No | 29 | VRT | 25 | NED | / | / |
| CTE005 | Mixed* | IA | 3 | 30% | No | Neg | SLN (3) | No | 10 | VRT + CHT | 19 | NED | / | / |
| CTE037 | CS | IA | 3 | 36% | No | Neg | SLN (3) | No | 16 | VRT + CHT | 29 | NED | / | / |
| CTE025 | Serous | IA | 3 | 42% | No | Neg | SLN (2) | No | 13 | VRT + CHT | 36 | NED | / | / |
| CTE003 | Serous | IIIA | 3 | 100% | Yes | Neg | SLN (11) | Yes** | 26 | CHT + EBRT + VRT | 32 | AWD | 21 (R) | Lung |
| CTE024 | CS | IIIC1 | 3 | 95% | Yes | Pos | SLN + R LND (2 +5) | No | 25 | CHT + EBRT + VRT | 36 | DOD | 18 (R) | Lung; Peritoneum; Pelvic, Para-aortic & Supraclavicular Nodes |
| CTE029 | Serous | IIIC1 | 3 | 95% | Yes | Pos | SLN (3) | No | 31 | CHT | 4 | DOD | 3 (PD) | Bone; Liver |
| CTE016 | Endom | IIIC2 | 1 | 72% | Yes | Pos | SLN + LND (3+34) | No | 94 | CHT + EBRT + VRT | 38 | NED | / | / |
| CTE033 | Serous | IIIC2 | 3 | 0% | No | Pos | LND (44) | No | 35 | CHT + EBRT + VRT | 32 | NED | / | / |
| CTE031 | Serous | IVB | 3 | 21% | No | Neg | SLN (3) | Yes | 19 | CHT | 29 | DOD | 13 (R) | Peritoneum; Omentum; Pelvic & Para-aortic nodes |

* Mixed histology: serous and endometrioid

** Uterine serosa. Abbreviations: AWD, alive with disease; CHT, chemotherapy; CS, carcinosarcoma; DOD, dead of disease; EBRT, external beam radiation therapy; Endom, endometrioid; LN, lymph node; LND, lymphadenectomy (number of nodes is how many lymph nodes were taken); NED, no evidence of disease; Neg, negative; Pos, positive; SLN, sentinel lymph node (number of nodes for SLN is how many SLN were taken); PD, progression of disease; VRT, vaginal brachytherapy.

## Somatic DNA junction in primary tumors

Fig 1 provides the schematic representation of monitoring ctDNA in plasma utilized in this study. Fresh frozen surgically resected primary tumor tissues were assessed for tumor purity by a board-certified pathologist and estimated to contain at least 60% tumor cellularity. Mate-Pair sequencing was performed on DNA isolated from the primary tumors to an average sequencing depth of 86 million fragments, with average bridged (junction spanning) coverage of 56X (S1 Table). Mate-Pair sequencing results are illustrated using Genome plots which provide a visual profile of somatic structural variants present in tumors. Chromosome coverage is colored according to their bioinformatically determined level; with grey, blue and red dots indicating normal diploid, gains and losses, respectively and junctions represented as black lines linking distal chromosomal regions. The genome plot for exemplar case CTE024 (Fig 2A) reveals extensive aneuploidy, with multiple chromosomal gains/losses. A large number of junctions were detected (238) with many interlinked complex rearrangements dispersed across the genome. DNA junctions were detected in all 11 endometrial tumors with an average of 174 (range 6–327) (Fig 2B and S1 Table). Genome plots of additional cases CTE025, CTE029 and CTE031 exemplify the unique chromosomal profiles present in different tumors (Fig 2C–2E). Additional genome plots are presented in S1 Fig. While commonalities in chromosomal gains and losses are clearly identifiable, no identical somatic DNA junctions were observed between cases.

## ctDNA screening in plasma

DNA junctions unique to each endometrial cancer were selected according to criteria outlined in the methods (S1–S3 Tables). Primers were designed to flank the junction breakpoints (S6 Table). Both pre- and post-surgical plasma was collected for all patients but one; CTE029, who had only post-surgical blood available (S4 Table). In case CTE029 primers were designed flanking a chromosome 3 gain junction, to generate an 89bp amplicon compatible with the fragment size of cfDNA in circulation (Fig 3Ai and 3Aii). Standard PCR confirmed the junction as somatic and specific to the tumor (Fig 3Aiii) and qPCR on the post-surgical plasma cfDNA positively detected a ctDNA specific peak equivalent to the tumor control (Fig 3Aiv) at a cycle threshold (Ct) of 33.264. Quantification predicted a ctDNA level of 0.31% of the total cfDNA (Fig 4, S7 Table). During the course of adjuvant treatment, this patient experienced systemic disease progression (liver and bone metastases) and died of disease four months post-surgery (Table 1).

Three junctions were selected for CTE025, a stage 1A endometrial cancer (S3 and S4 Tables). Primers designed for an intrachromosomal 17 junction hitting genes *ACACA* and *PLCD3* (Fig 3Bi) positively detected ctDNA in both pre- and post-surgical plasma cfDNA by qPCR (Ct values 34.797 and 33.841 respectively), with similar amplicon melting profile to primary tumor DNA, and no signal in negative controls (Fig 3Bii). Control gene amplicon NAGK qPCR generated expected positive signals; with no signal in the water (no template) control (Fig 3Biii). Primers for an additional chromosome 3–17 junction only detected ctDNA in pre-surgical plasma (Ct = 35.533), while a chromosome 3–9 junction was undetectable in both draws (S2 Fig) introducing questions of assay sensitivity or tumor heterogeneity (see Discussion). For stage IVB serous endometrial cancer CTE031, both interchromosomal 5–17 and intrachromosomal 13 junctions were negative by qPCR in pre- and post-surgical plasma (Fig 3Ci and 3Cii and S2 Fig). A higher sensitivity semi-nested PCR technique with extended amplification cycles confirmed junction 5–17 as undetectable (Fig 3Ciii).

Overall ctDNA was detected in 60% (6/10) pre-surgical plasma cfDNA (Fig 4). In 2 of the 5 stage IA endometrial cancers (40%), ctDNA was detected pre-surgery, with one patient (20%) demonstrating continued ctDNA presence 6 weeks post-surgery. In all 3 of the patients with

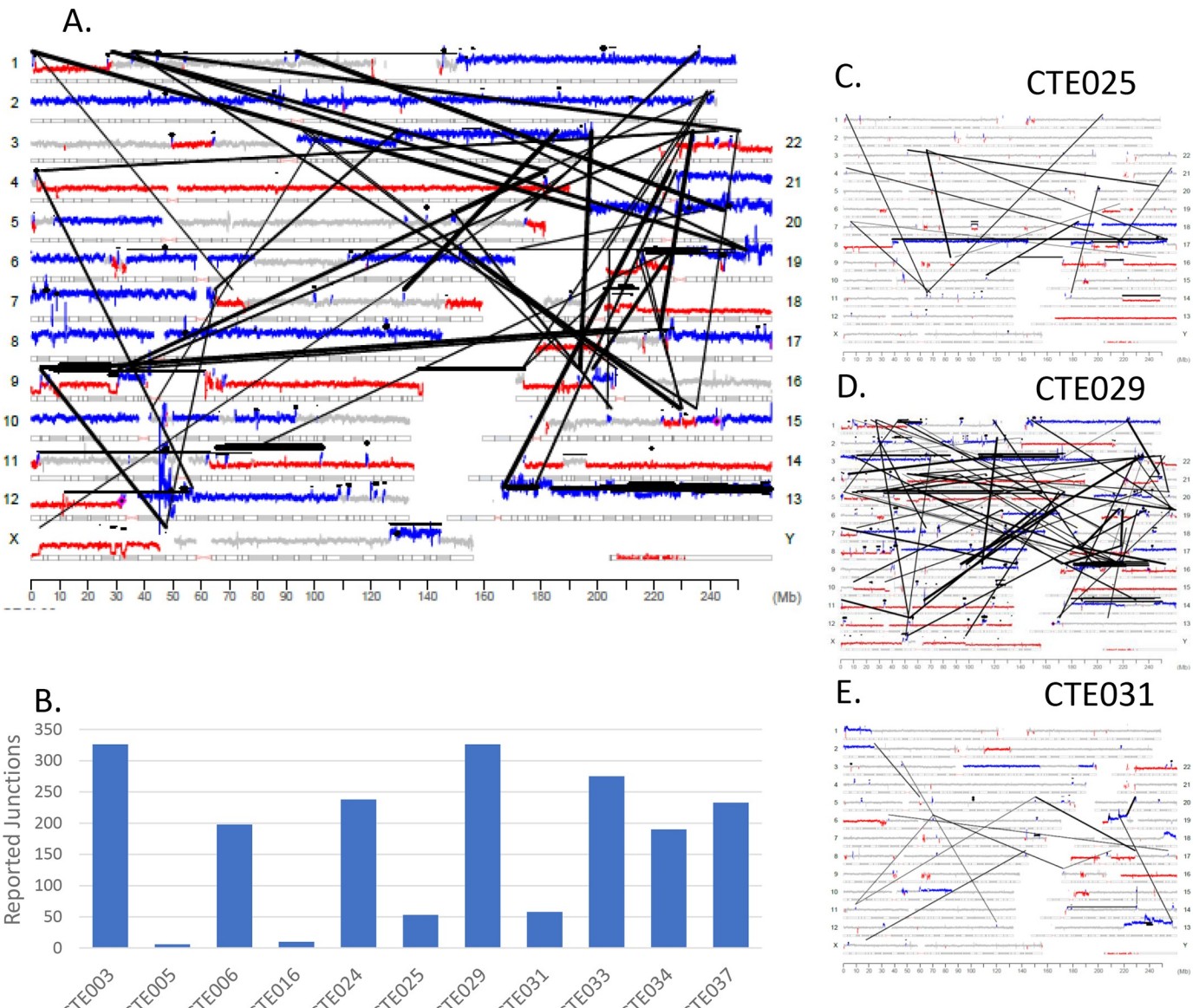

**Fig 2. Mate–Pair sequencing results from primary tumor DNA. (A)** Genome plot for primary tumor of case CTE024. Chromosomes are listed on the left and right Y–axis'; basepair position is on the X axis. Grey cytobands indicate genomic loci bands. The height of the dots each represents the average number of reads over 30 k bases. Grey color indicates wild–type 2 copy state of DNA, blue indicates gains, red indicates losses. Black dots indicate small intrachromosomal rearrangements, while the black lines indicate interchromosomal rearrangements or larger intrachromosomal junctions. **(B)** Bar graph showing the number of chromosomal junctions detected in the primary tumor of 11 cases with a threshold of at least 3 supporting reads per junction. **(C)** Genome plot for cases CTE025, **(D)** CTE029, and **(E)** CTE031.

lymphatic spread of disease and available pre-surgical plasma, ctDNA was detected in pre-surgical plasma. In 2 of these patients ctDNA was not detected post-surgically, both of which received bilateral pelvic and para-aortic lymphadenectomy. Indeed, in the two patients with positive nodes that underwent sentinel lymph node (SLN) biopsy (for one we had only post-surgical plasma available), ctDNA was detected post-operatively. In one patient with stage IVB disease, ctDNA was not detected either pre- or post-surgery.

Generally, the detection of ctDNA pre-surgery was consistent with clinical indicators of aggressive disease such as advanced stage (4/5–80%), serous histology (4/5–80%), deep

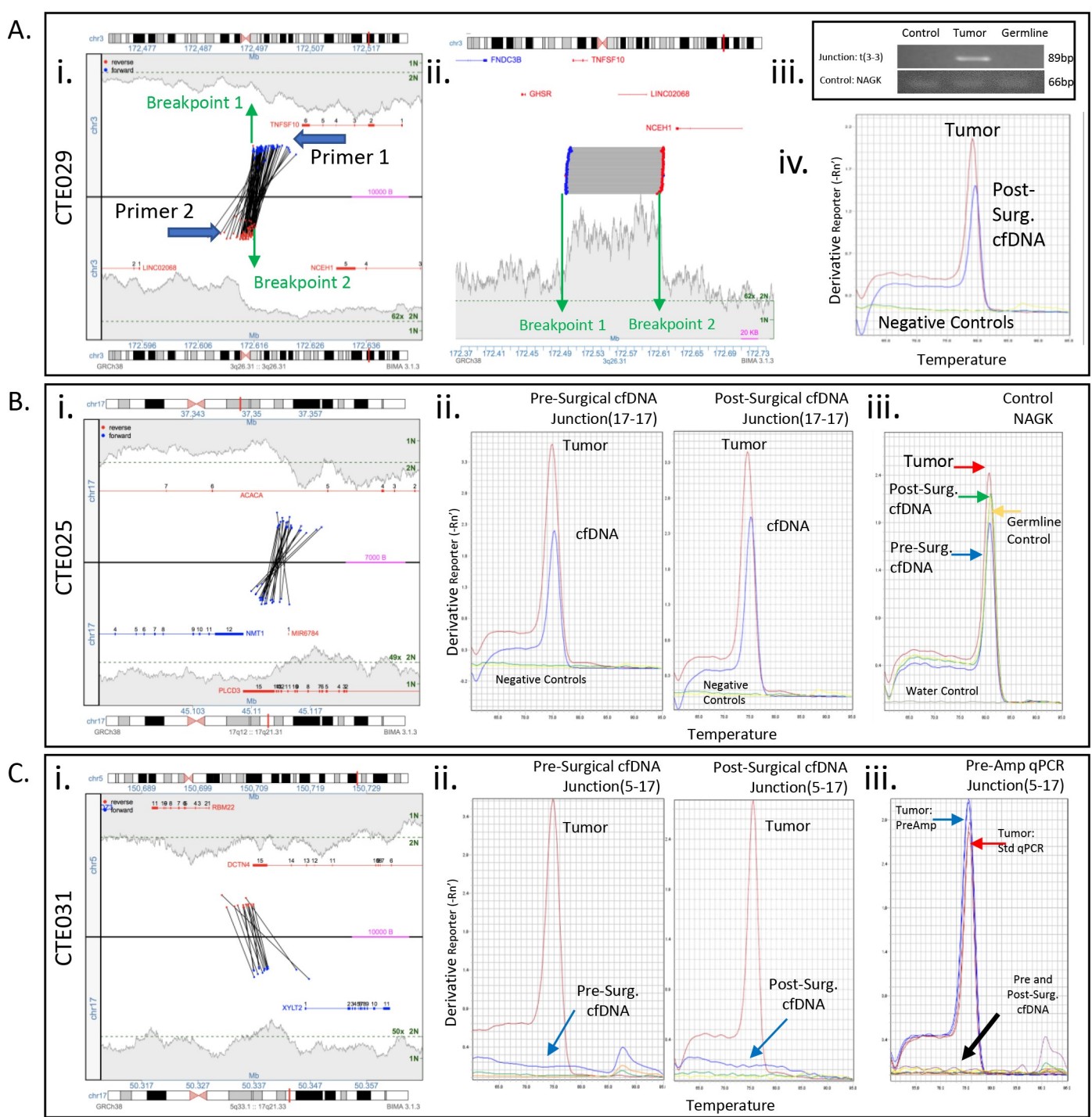

**Fig 3. Junction primer design and qPCR cfDNA detection results. (A)** Case CTE029. *Left Panel*: Junction plots of selected intrachromosomal junction of chromosome 3 with base positions 172MB–173MB (3a–3b). The middle line separates the 2 chromosomal areas involved in the junction. The lines across show the fragments that span the junction and support the rearrangement. The position of each read of a supporting fragment is located by a dot and color coded by strand, red for reads mapping to the reverse strand, blue for forward strand. The bridged coverage for the region is illustrated by the shaded area. The green dotted line on the y–axis indicates the bridged coverage averaged across the entire genome (normalized to estimate 2N and 1N). Genes within the region are displayed, indicating exon location and strand direction. General relative positions and directions of designed primers indicated. *Middle panel*: Linear plot of junction. *Right Panel*: Standard endpoint PCR of junction 3a–3b and amplification control product NAGK in pooled genomic DNA, patient tumor, and patient PBMCs. Sybr green qPCR amplicon melting curve for 3a–3b junction in patient 6–week post–surgical cfDNA. **(B)** Case CTE025. *Left Panel*: Junction plot of selected junction. *Middle Panels*: Sybr green qPCR melting curve of amplicons of 17a–17b junction in patient pre–and post–surgical cfDNA. *Right Panel*: qPCR melting curve for control product NAGK. **(C)** Case CTE031. *Left Panel*: Junction plot of selected junction. *Middle Panels*: Sybr green qPCR melting curve of amplicons of 5–17 junction in patient pre–and post–surgical cfDNA. *Right Panel*: qPCR melting curve for junction 5–17 in presurgical cfDNA following 30 cycles of pre–amplification.

| | CTE006 | CTE005 | CTE037 | CTE031 | CTE034 | CTE025 | CTE003 | CTE024 | CTE016 | CTE033 | CTE029 |
|---|---|---|---|---|---|---|---|---|---|---|---|
| Stage | IA | IA | IA | IVB | IA | IA | IIIA | IIIC | IIIC | IIIC | IIIC |
| Myometrial Invasion | <25% | 25-50% | 25-50% | <25% | 25-50% | 25-50% | >75% | >75% | 50-75% | <25% | >75% |
| LVSI | Pos | Neg | Neg | Neg | Neg | Neg | Pos | Pos | Pos | Neg | Pos |
| Lymph Node Status | Neg | Neg | Neg | Neg | Neg | Neg | Neg | Pos | Pos | Pos | Pos |
| Lymph Node Assessment | SLN | SLN | SLN | SLN | SLN | SLN | SLN | UP LND | P+PA LND | P+PA LND | SLN |
| Recurrence or Progression | Yes | No | No | Yes | No | No | Yes | Yes | No | No | Yes |
| | | | | | | | | | | | |
| Histology | E | Mx | CS | S | E | S | S | CS | E | S | S |
| | | | | | | | | | | | |
| ctDNA Pre-Surgical | ND | ND | ND | ND | D | D | D | D | D | D | NA |
| ctDNA Post-Surgical | ND | ND | ND | ND | ND | D | ND | D | ND | ND | D |
| | | | | | | | | | | | |
| Junctions Assessed in Plasma (N) | 1 | 1 | 1 | 2 | 2 | 3 | 1 | 2 | 1 | 1 | 1 |
| %ctDNA Pre-Surgical | / | / | / | / | 1.69 | 0.05 | 0.08 | 0.19 | 0.03 | 0.38 | NA |
| %ctDNA Post-Surgical | / | / | / | / | / | 0.2 | / | 0.16 | / | / | 0.31 |

**Fig 4. Heat map characterizing and correlating clinical findings to the presence or absence of detected circulating tumor DNA (ctDNA).** Abbreviations: CS, carcinosarcoma; D, detected; E, endometrioid; G, grade; LVSI, lymphovascular space invasion; Mx, Mixed; NA, not available; ND, not–detected; Neg, negative; P + PA LND, pelvic + para–aortic lymphadenectomy; Pos, Positive; S, Serous; UP LND, unilateral pelvic lymphadenectomy.

myometrial invasion (3/3–100%), lymphatic spread of disease (3/3–100%), and lympho-vascular space invasion (3/4–75%) (Fig 4).

In the 6-week post-surgical plasmas, ctDNA was detected in 27% (3/11) of cases (Fig 4). Two of the six cases with positive pre-surgical ctDNA were also positive post-surgically. The third post-surgical positive case, CTE029, had no pre-surgical blood draw specimen available. All patients in which ctDNA was detected post-surgically had type II endometrial cancer. Two of 3 patients with post-surgery ctDNA detection experienced recurrence or progression of disease. Overall, in 3 out of 5 patients (60%) who experienced recurrence or progression of disease, circulating tumor DNA was detected either pre- or post-surgery (Fig 4).

## Discussion

No reliable blood-based biomarkers are available for patients with endometrial cancer. In many solid cancers, ctDNA is emerging as a promising non-invasive marker. In this pilot study, we demonstrate the feasibility of using personalized tumor specific junctions for measuring ctDNA burden in endometrial cancer patient's plasma. These preliminary results suggest a tentative relationship between pre-surgical ctDNA detection and the presence of a cancer with aggressive tumor characteristics or advanced stage.

To our knowledge, only a few studies have reported data on ctDNA in endometrial cancer [11–16, 24]. CtDNA was detected preoperatively in 15% (10/68 patients) to 41% (21/51 patients) of patients with endometrial cancer, using tumor-associated mutations [12–14]. Using tumor specific junctions, we detected ctDNA in 60% (6/10) of patients with high grade or advanced stage endometrial cancer pre-surgery. Consistent with previously reported studies [12–14], we generally found pre-surgical ctDNA detection to correspond to clinical indications of more aggressive disease, such as lymphatic spread of disease, deep myometrial invasion, advanced stage, lympho-vascular space invasion and type II histology (Table 1 and Fig 4). In our series, all 3 patients with documented lymphatic spread of disease had ctDNA detected in pre-surgical plasma. If replicated in a larger population utilizing DNA from preoperative biopsies, this information may possibly help surgeons to identify patients with occult lymphatic dissemination in the preoperative setting. Meanwhile, CTE031, a stage IVB tumor with serous histology, had undetectable ctDNA both pre- and post-surgery. It is possible that the lack of detection of ctDNA could be related to the peritoneal spread and absence of lymphatic or hematogenous spread in this patient. In a study set with 3 stage IV endometrial cancer patients, only 1 had measurable ctDNA [13]. However, whether this lack of detection is due to insensitivities of ctDNA assays, or biological mechanisms controlling the release of ctDNA, remains to be determined.

We were able to detect pre-surgical ctDNA in 40% (2/5) of patients with stage IA endometrial cancer. This is similar to the ctDNA detection of 20% (8/35 patients) to 34% (12/35 patients) of early stage endometrial cancer reported previously [12, 14]. One of these cases, CTE025 with serous histology demonstrated ctDNA 6-weeks post-surgery, and 36 months later there was no clinical evidence of disease (Table 1 and Fig 4). However, she underwent adjuvant chemotherapy and vaginal brachytherapy. Therefore, the significance of ctDNA in the blood of early stage endometrial cancer and any correlations between the presence of ctDNA pre-surgery and long-term outcomes warrants larger extended studies. CtDNA qPCR assays with CTE025 produced variable results based on which junction was tested (Fig 3B and S3 Fig) which could stem from tumor heterogeneity. However, as these three junctions are believed related to a single chromplectic event, the ctDNA levels may be at the lower thresholds of primer sensitivities, with inconsistent shedding of tumor genomic DNA to the blood [25].

Three cases showed continued ctDNA six weeks post-surgery. One serous patient with lymphatic spread of disease (CTE029) experienced disease progression during the course of adjuvant treatment (Table 1). The second, a carcinosarcoma patient with lymphatic spread of disease (CTE024) experienced a disease recurrence 18 months post-surgery and died of disease 18 months later (Table 1). The third is a stage I serous endometrial cancer patient (CTE025) who showed no relapse at 36 months of follow-up. However, the administration of adjuvant therapy following surgery makes it difficult to draw conclusions in defining long-term outcomes (Table 1). A few studies describe the possible clinical utility of ctDNA testing in the post-operative setting and follow-up of patients with endometrial cancer[11, 15, 16]. Pereira et al. found post-treatment detection of ctDNA was related to worst survival in 10 patients with gynecological cancers [11]. However, conclusions are limited as only two endometrial cancer patients were included in their analysis.

Although this pilot study effectively detected ctDNA in the majority of patients, inter-case variability of selected junctions indicates greater sensitivity is required to detect lower ctDNA burden. While fragment size is limited, semi-nested PCR assays were applied to increase sensitivity when ctDNA was undetected. Reduced specificity was a concern with increased PCR cycles, but significantly no additional on target signal was detected in negative cases. One limitation of this study was the application of single junction assays in the method development

stage for several cases. Application of a larger junction panel would increase sensitivity of detection, although limited plasma volumes would demand multiplexing to ensure adequate sample input into PCR reactions. The personalized aspect of patient specific DNA panels also challenges our current processes of laboratory-based testing protocols, being more labor intensive upfront in panel design with limited assay validation. However, the tumor heterogeneity found in endometrial cancer currently evades a generic tumor assay for cfDNA screening and the real-time benefit of patient plasma monitoring demands more high-sensitivity assays. Potentially, an individualized junction assay, as presented here, could serve as a reflex test for those patients for whom a mutation-based panel is not suitable.

The small sample size and limited follow-up time of this pilot study precludes significant clinical interpretation. However, this study adds to the limited data published on ctDNA in endometrial cancer. The ability to preoperatively identify patients with occult lymphatic dissemination through detectable ctDNA levels could benefit endometrial patients, although a larger cohort of patients is required to confirm these findings. The clinical significance of detectable ctDNA at 6 weeks post-surgery is still unclear but merits further study, particularly in the setting of adjuvant therapy and follow up to monitor tumor burden, response to therapy and recurrence.

In conclusion, this pilot study effectively demonstrates the feasibility of using personalized tumor specific junction panels for detecting ctDNA in the plasma of endometrial cancer patients. Despite a limited sample size, the results suggest a relationship between pre-surgical ctDNA detection and the presence of a cancer with aggressive tumor characteristics or advanced stage, though a larger study is needed to confirm these findings.

## Supporting information

**S1 Fig. Genome plots of additional cases.** Chromosomes are listed on the left and right Y-axis'; basepair position is on the X axis. Grey cytobands indicate genomic loci bands. The height of the dots each represents the average number of reads over 30 k bases. Grey color indicates wild-type 2 copy state of DNA, blue indicates gains, red indicates losses. Black dots indicate small intrachromosomal rearrangements, while the black lines indicate interchromosomal rearrangements or larger intrachromosomal junctions.
(TIF)

**S2 Fig. Serial dilutions of tumor DNA.** qPCR standard curves produced by Applied Biosystems ViiA-7 generated for each junctions primer pair tested in cfDNA. Y-axis: Ct, X-axis quantity of input (ng/ul).
(TIF)

**S3 Fig. Case CTE025.** (A) *Left Panel*: Junction plot of selected junction. *Right Panel*: Sybr green qPCR melting curve of amplicons of 3–17 junction in patient pre- and post-surgical cfDNA. (B) (A) *Left Panel*: Junction plot of selected junction. *Right Panel*: Sybr green qPCR melting curve of amplicons of 3–9 junction in patient pre- and post-surgical cfDNA. (C) qPCR CT values for each junction
(TIF)

**S1 Table. Mate-pair sequencing quality control data.**
(DOCX)

**S2 Table. List of junctions.** All junctions detected in each case using the BIMA algorithm with at least 3 supporting reads following mate pair sequencing. Number of associates: Number of mate-pair reads supporting the presence of the indicated junction. GeneA, GeneB: Gene

interrupted by junction on either region.
(XLSX)

**S3 Table. List of mate-pair reads.** List of all supporting reads for each junction in all cases that have at least 3 supporting reads. Number of associates: Number of mate-pair reads supporting the presence of the indicated junction. GeneA, GeneB: Gene interrupted by junction on either region.
(XLSX)

**S4 Table. Extraction details, number of junctions assessed, and qPCR reaction input.**
(XLSX)

**S5 Table. List of junctions selected for ctDNA monitoring.** Number of associates: number of mate-pair reads supporting the presence of the indicated junction. GeneA, GeneB: Gene interrupted by junction on either region.
(XLSX)

**S6 Table. List of primers and amplicons sequences used for junction detection in plasma.** Semi-nested PCR was performed with the indicated primers on total cfDNA for which ctDNA was undetected pre-surgically.
(XLSX)

**S7 Table. qPCR derived Ct's for ctDNA positive cases used in ctDNA% calculation.**
(XLSX)

**S8 Table. Patient demographics.** CHT, chemotherapy; EBRT, external beam radiation therapy; LND, lymphadenectomy; SLN, sentinel lymph node; VBT, vaginal brachytherapy.
(DOCX)

## Acknowledgments

We are grateful to the patients who participated in this study.

## Author Contributions

**Conceptualization:** Tommaso Grassi, Faye R. Harris, Stephen J. Murphy, George Vasmatzis, Andrea Mariani.

**Data curation:** Tommaso Grassi, Faye R. Harris, James B. Smadbeck, Stephen J. Murphy, Francesco Multinu, Piyan Zhang, Giannoula Karagouga, Serena Cappuccio, John C. Cheville, George Vasmatzis, Andrea Mariani.

**Formal analysis:** Tommaso Grassi, Faye R. Harris, James B. Smadbeck, Stephen J. Murphy, Serena Cappuccio, George Vasmatzis, Andrea Mariani.

**Funding acquisition:** George Vasmatzis, Andrea Mariani.

**Investigation:** Tommaso Grassi, Faye R. Harris, James B. Smadbeck, Stephen J. Murphy, Matthew S. Block, Francesco Multinu, Minetta C. Liu, Alyssa Larish, Serena Cappuccio, John C. Cheville, George Vasmatzis, Andrea Mariani.

**Methodology:** Tommaso Grassi, Faye R. Harris, James B. Smadbeck, Stephen J. Murphy, Francesco Multinu, Piyan Zhang, Giannoula Karagouga, Minetta C. Liu, Serena Cappuccio, John C. Cheville, George Vasmatzis, Andrea Mariani.

**Project administration:** Janet L. Schaefer Klein, Alyssa Larish, Maureen A. Lemens, Marla Kay S. Sommerfield, George Vasmatzis, Andrea Mariani.

**Supervision:** Stephen J. Murphy, Minetta C. Liu, George Vasmatzis, Andrea Mariani.

**Writing – original draft:** Tommaso Grassi, Faye R. Harris, Stephen J. Murphy.

**Writing – review & editing:** Tommaso Grassi, Faye R. Harris, James B. Smadbeck, Stephen J. Murphy, Matthew S. Block, Francesco Multinu, Janet L. Schaefer Klein, Piyan Zhang, Giannoula Karagouga, Minetta C. Liu, Alyssa Larish, Maureen A. Lemens, Marla Kay S. Sommerfield, Serena Cappuccio, John C. Cheville, George Vasmatzis, Andrea Mariani.

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
