## [Decision Letter · Decision Letter 0]

24 Mar 2021

PONE-D-21-07462

Personalized tumor-specific DNA junctions to detect circulating tumor in patients with endometrial cancer

PLOS ONE

Dear Dr. Grassi,

Thank you for submitting your manuscript to PLOS ONE. After careful consideration, we feel that it has merit but does not fully meet PLOS ONE’s publication criteria as it currently stands. Therefore, we invite you to submit a revised version of the manuscript that addresses the points raised during the review process.

We look forward to receiving your revised manuscript.

Kind regards,

Alvaro Galli

Academic Editor

PLOS ONE

Journal Requirements:

2. Please note that PLOS ONE has no limit on word count. Therefore, we recommend that details of methods required for another researcher to reproduce the experiments be included in the main manuscript Methods rather than Supporting Information.

3. Thank you for including your ethics statement:  "This prospectively collected study was approved by the Mayo Clinic Institutional Review Board under IRB# 15-005545.".  

Please provide additional details regarding participant consent. In the ethics statement in the Methods and online submission information, please ensure that you have specified (i) whether consent was informed and (ii) what type you obtained (for instance, written or verbal, and if verbal, how it was documented and witnessed). If your study included minors, state whether you obtained consent from parents or guardians. If the need for consent was waived by the ethics committee, please include this information.

Reviewers' comments:

Reviewer's Responses to Questions

**Comments to the Author**

1. Is the manuscript technically sound, and do the data support the conclusions?

Reviewer #1: Yes

Reviewer #2: Yes

2. Has the statistical analysis been performed appropriately and rigorously? 

Reviewer #1: Yes

Reviewer #2: N/A

3. Have the authors made all data underlying the findings in their manuscript fully available?

Reviewer #1: No

Reviewer #2: No

4. Is the manuscript presented in an intelligible fashion and written in standard English?

Reviewer #1: Yes

Reviewer #2: Yes

5. Review Comments to the Author

Reviewer #1: In their manuscript, “Personalized tumor-specific DNA junctions to detect circulating tumor in patients with endometrial cancer,” Grassi et al. use mate-pair sequencing of tumor DNA in 11 patients with endometrial cancer to identify chromosomal rearrangements. They then design personalized qPCR assays for circulating tumor DNA (ctDNA) analysis of pre-treatment and post-surgery plasma. They find that detection of ctDNA pre-surgery appears to be associated with more aggressive disease, and 2/3 patients with post-surgery ctDNA detected developed disease recurrence. Although the findings will need to be validated in larger studies, the experimental methods are sound, and the manuscript is well-written. I would appreciate some additional technical details and underlying data included the published manuscript:

1. The amount of cell-free DNA (cfDNA) used for analysis will affect the sensitivity of the assay, and it is unclear how much cfDNA was used for each sample. It looks like 1/10 of the total cfDNA extracted from each sample? It would be helpful to include this in supplementary table 5.

2. The authors assessed sensitivity, specificity, and robustness of each qPCR assay using diluted primary tumor DNA. It would be helpful to include this information as a supplement for each primer set. What is the limit of detection based on the dilution series? How was robustness measured? What sensitivity, specificity, and robustness was required for each primer combination?

3. It would be helpful to include the qPCR threshold cycles for each sample that was used for calculation of ctDNA fraction in the supplement of the manuscript.

4. The supplementary methods say that 1-5 unique junctions were selected for plasma screening for each patient. However, Supplementary Table S4 has a max of 3 junctions per patient. Are some of the primer sequences missing?

5. The number of junctions per patient will affect sensitivity of ctDNA detection. It would be helpful to include this in Figure 4.

Reviewer #2: Levels of circulating ctDNA have been associated with prognosis in several types of cancer, and are generally associated with treatment response and overall tumor burden. ctDNA levels are frequently assessed by direct DNA sequencing assays that calculate the variant allele frequency of mutations specific to tumors as a fraction of the total number of reads present at a given locus. In this study, the investigators evaluated the performance of customized junction probes designed to span structural variations to monitor ctDNA levels in the bloodstream of patients with endometrial cancer. This study harbors significant similarity to the Harris et al 2016 ovarian cancer study published by many of the same investigators.

The writing is clear and the methods and observations are described with sufficient detail. It is reasonable to ctDNA levels in patients with different prognostic assessments would demonstrate that high ctDNA levels are associated aggressive histology, but with this number of samples, in the absence of any statistical analysis, no clear association can be clearly inferred. The language in the abstract and discussion is suitable circumspect, although in the absence of a sufficiently powered experiment this work is a proof of concept.

Main points:

1) What is the motivation for using a personalized assay targeted at structural variation junctions unique to a given patient, as opposed to a strategy that deeply sequences the coding space of a small number of frequently mutated in endometrial cancer? As a practical matter, the bioinformatic and operational burden is high for this approach. It appears from a look at cBioPortal that there are a number of frequently mutated driver genes (including TP53, PIK3CA, ARID1A, PTEN, PIK3R1, KRAS, etc.) that would be suitable targets for a panel-based screening method.

2) Was any orthogonal assessment of ctDNA quantity performed? This would be important both to compare the accuracy and consistency of the qPCR-based assessment to a conventional assessment based on the presence of mutations or copy number.

3) The MP data should be desposited in a suitable public repository e.g. the EGA.

Minor notes:

1) The representation of junctions in Figure 2A is somewhat unconventional, as structural variations frequently drawn using Circos plots. It is unclear what additional information is provided by panels 2C, 2D, and 2E.

6. PLOS authors have the option to publish the peer review history of their article (what does this mean?). If published, this will include your full peer review and any attached files.

Reviewer #1: No

Reviewer #2: No

---

## [Author Response · Author response to Decision Letter 0]

8 May 2021

Responses to comments of Reviewer 1:

1) The amount of cell-free DNA (cfDNA) used for analysis will affect the sensitivity of the assay, and it is unclear how much cfDNA was used for each sample. It looks like 1/10 of the total cfDNA extracted from each sample? It would be helpful to include this in supplementary table 5.

In a clinical testing scenario DNA inputs into assays would be maximized to facilitate optimal sensitivity. However, in this pilot protocol development assay, input cfDNA levels were restricted to lower 1/10th volumes of total extracts to preserve the DNA for potential experimental repeats, avoiding complete depletion of the precious clinical specimens. As recommended, the cfDNA input has been added to supplementary table 4.

2) The authors assessed sensitivity, specificity, and robustness of each qPCR assay using diluted primary tumor DNA. It would be helpful to include this information as a supplement for each primer set. What is the limit of detection based on the dilution series? How was robustness measured? What sensitivity, specificity, and robustness was required for each primer combination?

As suggested, all available curves for all the primer sets are now included as supplement figure 2. Specificity, sensitivity, and robustness were assessed by generating standard curves from serial dilutions of tumor DNA. Similarly, as suggested by the reviewer, limit of detection can also be determined by the dilution series. To illustrate this process, we include three examples from two cases below, where we determined the junction primer set limit of detection. For both primer sets for of CTE031, the limit of detection was 64-129pg/ul with a 3ul input, a range based on the concentration of the last stable dilution detection to the first dilution where one of the replicates was undetected. For CTE033 14-17, the limit of detection was 4-8pg/ul with a 3ul input. [figure included with "Response to Reviewers.doc"]

For a subset of the primer sets we did not extend the dilutions all the way to “undetected” and only used them to define linearity. An R^2 of at least 0.90 was required for sufficient robustness for each primer set to be used on cfDNA. Robustness was assessed by closeness of duplicates. Specificity was determined by testing the primers against pooled human genomic DNA (Promega, #G3041) and prepared pooled normal cell-free DNA. 100% specificity was required for primers to be selected and tested on cfDNA. No significant amplicons were acceptable in negative control DNA or water that demonstrated similar melting temperatures to the specific product derived from the tumor DNA.

3) It would be helpful to include the qPCR threshold cycles for each sample that was used for calculation of ctDNA fraction in the supplement of the manuscript.

This information has been added to supplementary table S7.

4) The supplementary methods say that 1-5 unique junctions were selected for plasma screening for each patient. However, Supplementary Table S4 has a max of 3 junctions per patient. Are some of the primer sequences missing?

To address this inconsistency, we have added additional information in the methods and supplementary material for clarification. For each case, all detected junctions were reviewed bioinformatically, and up to 5 unique junctions were selected for further workup. However, after initial testing on tumor DNA, a number of these junctions failed selection criteria prior to testing on cfDNA due to poor primer specificity or an R^2 <0.90. Clarifying information has been added to the manuscript and the number of junctions initially selected and the number of junctions tested on cfDNA are listed in supplementary table 4. In addition, the full list of junctions detected and considered for selection are now included in supplementary table 2, and all supporting reads for those junctions are now in supplementary table 3. 

5) The number of junctions per patient will affect sensitivity of ctDNA detection. It would be helpful to include this in Figure 4. 

As suggested, this information has been added to Figure 4

Responses to comments of Reviewer 2:

1) What is the motivation for using a personalized assay targeted at structural variation junctions unique to a given patient, as opposed to a strategy that deeply sequences the coding space of a small number of frequently mutated in endometrial cancer? As a practical matter, the bioinformatic and operational burden is high for this approach. It appears from a look at cBioPortal that there are a number of frequently mutated driver genes (including TP53, PIK3CA, ARID1A, PTEN, PIK3R1, KRAS, etc.) that would be suitable targets for a panel-based screening method.

The primary reasons for choosing a junctions-based personalized assay are the following: 

I. We always found junctions in serous endometrial cancer, raising the theoretical upper limit of the diagnostic sensitivity to 100%. 

II. The assay, once developed, is very inexpensive and easily implemented because it is based on PCR and not sequencing.

III. The specificity and sensitivity of junctions is superior to mutations because there is no wild-type fragment containing the junction. 

The potential alternative strategy, suggested by the reviewer, to deeply sequence the coding space of a small number of frequently mutated genes has the following limitations:

I. Sequencing of every serial sample could be more expensive than a simple PCR assay and more cumbersome to implement.

II. It is not certain that mutations of the targeted genes will be found in all patients’ tumors. 

III. Even a gene like TP53 that is almost always deactivated in serous endometrial cancer is not always mutated as it can be deactivated by junctions and deletions. 

Our approach, while seemingly difficult bioinformatically and operationally, has been reduced to only a few hours of data analysis and primer design. Our pipelines to detect junctions from whole genome data are automated and have been used in numerous publications. 

Our aim here was to show the viability of using personalized junctions for the detection of circulating tumor DNA in endometrial cancer. Junctions provide a desirable monitoring target in a personalized ctDNA detection assay as you can design primers for a tumor-specific amplicon. This reduces the chance of false positive signal and produces a simpler and cheaper assay. Additionally, a panel of mutations will often find only one or even zero monitorable mutations for a given sample. For example, using cBioportal and taking any gene with mutations in at least 10% of the 197 endometrial cases we find that 59/197 (30%) cases do not have a mutation in any of the genes and 103/197 (52%) only have a mutation in one of the genes. Therefore, one could also use a personalized assay like the one assessed in this manuscript as reflex test if a mutation-panel fails. This would allow for the monitoring of all patients with multiple probes to improve robustness of the process. We thank the reviewer for the comment, and we clarified this point in the conclusions.

2) Was any orthogonal assessment of ctDNA quantity performed? This would be important both to compare the accuracy and consistency of the qPCR-based assessment to a conventional assessment based on the presence of mutations or copy number.

This is a good point and would have been valuable to produce such data, but the limited allowance of funds and available plasma precluded orthogonal assessments in this study. However, we found in other studies that copy number assessment was inferior to PCR for detection of recurrence due to poor sensitivity of the sequencing techniques. We therefore opted towards PCR early in our initiative. With respect to mutations, we also have concluded that detecting junctions by PCR is more specific, and more sensitive, than sequencing mutations as we explained above. We think that our assay is relatively inexpensive, innovative and with wider applicability, and in some cases, it might be the only possible assay to use.

3) The MP data should be deposited in a suitable public repository e.g. the EGA.

For this specific study, our current IRB protocol and consent form did not allow us to deposit genomic data to a public repository because it is considered health information, and we did not get the explicit permission from the patients to do so. We are looking into a study modification which may allow us to deposit the data, however, the possibility of re-consenting the patients presents considerable difficulty. To make some of the MPseq data available without compromising patient privacy concerns, we have added 2 additional supplemental tables: We have added Supplementary Table 2 which includes a list of all junctions detected for each case (discordant genome as compared to reference genome), and Supplementary Table 3 which includes a report of all supporting reads for each junction. Note, data supporting concordant sequencing (which aligns with the reference genome and may contain low coverage information on SNPs) is not being included at this time. Investigators who would like access to this data are invited to email the corresponding author with a data request, which will be considered on a case-by-case basis in accordance with our IRB and general Mayo Clinic policies regarding sharing data. 

Please note, the number of total reported junction for a subpopulation of cases changed slightly from the previously submitted version. This is due to the data being re-run on the latest versions of our algorithmic pipelines for calling somatic structural variants in cancer genomes to ensure the robust reporting of data. The continual upgrading and refinement of our algorithmic pipeline provides progressive improvements in our masking of false positive and germline junctions, while improved mapping functions provide further resolution of true positive junctions in highly repetitive regions of the reference genome. To incorporate these improvements to our algorithms into the manuscript, we have revised the number of total junctions detected for each case in Figure 2 and supplementary Table 1.

4) The representation of junctions in Figure 2A is somewhat unconventional, as structural variations frequently drawn using Circos plots. It is unclear what additional information is provided by panels 2C, 2D, and 2E.

The genome U-plots are preferred to Circos plots here because they were shown to be superior in visualizing structural variants [Gaitatzes et al 2018]. The U-plots use the 2D space 3.5 times better that Circos plots making it easier to show breakpoint locations and structural variants (deletions, gains amplifications etc.) while additionally illustrating breakpoint orientations across the Genome. We routinely use these plots to demonstrate the variability seen in the junctions in terms of how many, and where along the genome they are seen. 

Gaitatzes A, Johnson SH, Smadbeck JB, Vasmatzis G. Genome U-Plot: a whole genome visualization. Bioinformatics. 2018;34(10):1629-34. 

Freitag CE, et. al: Assesment of isochromosome 12p and 12p abnormalities in germ cell tumors using fluorescence in situ hybridization, single-nucleotide polymorphism arrays, and next-generation sequencing/mate-pair sequencing. Human Pathology. 2021; 112: 20-34. 

Murphy SJ, et. al: Using genomics to differentiate multiple primaries from metastatic lung cancer. Journal of Thoracic Oncology. 2019; 14(9):1567-1582.

---

## [Decision Letter · Decision Letter 1]

17 May 2021

Personalized tumor-specific DNA junctions to detect circulating tumor in patients with endometrial cancer

PONE-D-21-07462R1

Dear Dr. Grassi,

We’re pleased to inform you that your manuscript has been judged scientifically suitable for publication and will be formally accepted for publication once it meets all outstanding technical requirements.

Kind regards,

Alvaro Galli

Academic Editor

PLOS ONE

Additional Editor Comments (optional):

Reviewers' comments:

Reviewer's Responses to Questions

**Comments to the Author**

1. If the authors have adequately addressed your comments raised in a previous round of review and you feel that this manuscript is now acceptable for publication, you may indicate that here to bypass the “Comments to the Author” section, enter your conflict of interest statement in the “Confidential to Editor” section, and submit your "Accept" recommendation.

Reviewer #1: All comments have been addressed

Reviewer #2: All comments have been addressed

2. Is the manuscript technically sound, and do the data support the conclusions?

Reviewer #1: Yes

Reviewer #2: Yes

3. Has the statistical analysis been performed appropriately and rigorously? 

Reviewer #1: Yes

Reviewer #2: Yes

4. Have the authors made all data underlying the findings in their manuscript fully available?

Reviewer #1: Yes

Reviewer #2: Yes

5. Is the manuscript presented in an intelligible fashion and written in standard English?

Reviewer #1: Yes

Reviewer #2: Yes

6. Review Comments to the Author

Reviewer #1: (No Response)

Reviewer #2: (No Response)

7. PLOS authors have the option to publish the peer review history of their article (what does this mean?). If published, this will include your full peer review and any attached files.

Reviewer #1: No

Reviewer #2: No

---

## [Editor Report · Acceptance letter]

1 Jun 2021

PONE-D-21-07462R1 

Personalized tumor-specific DNA junctions to detect circulating tumor in patients with endometrial cancer 

Dear Dr. Grassi:

I'm pleased to inform you that your manuscript has been deemed suitable for publication in PLOS ONE. Congratulations! Your manuscript is now with our production department. 

Kind regards, 

on behalf of

Dr. Alvaro Galli 

Academic Editor

PLOS ONE